# Electrical Stimulation Induces Retinal Müller Cell Proliferation and Their Progenitor Cell Potential

**DOI:** 10.3390/cells9030781

**Published:** 2020-03-23

**Authors:** Sam Enayati, Karen Chang, Hamida Achour, Kin-Sang Cho, Fuyi Xu, Shuai Guo, Katarina Z. Enayati, Jia Xie, Eric Zhao, Tytteli Turunen, Amer Sehic, Lu Lu, Tor Paaske Utheim, Dong Feng Chen

**Affiliations:** 1Schepens Eye Research Institute of Massachusetts Eye and Ear, Department of Ophthalmology, Harvard Medical School, Boston, MA 02114, USA; samenayati@gmail.com (S.E.); karen_chang@meei.harvard.edu (K.C.); ach.hamida@gmail.com (H.A.); kinsang_cho@meei.harvard.edu (K.-S.C.); james.shuaig@gmail.com (S.G.); k.zihlavnikova@gmail.com (K.Z.E.); x.jessica@outlook.com (J.X.); eric_zhao@brown.edu (E.Z.); tutteli_turunen@meei.harvard.edu (T.T.); utheim2@gmail.com (T.P.U.); 2Department of Medical Biochemistry, Oslo University Hospital, 0372 Oslo, Norway; 3Department of Ophthalmology, Drammen Hospital, Vestre Viken Hospital Trust, 3004 Drammen, Norway; 4Institute of clinical medicine, University of Oslo, 0318 Oslo, Norway; 5Department of Genetics, Genomics and Informatics, University of Tennessee Health Science Center, Memphis, TN 38163, USA; fxu10@uthsc.edu (F.X.); llu@uthsc.edu (L.L.); 6Department of Oral Biology; Faculty of Dentistry, University of Oslo, 0372 Oslo, Norway; amer.sehic@odont.uio.no; 7Department of Plastic and Reconstructive Surgery, Oslo University Hospital, 0027 Oslo, Norway

**Keywords:** electrical-stimulation, retina, glial cells, Müller cells, proliferation, retinitis pigmentosa

## Abstract

Non-invasive electrical stimulation (ES) is increasingly applied to improve vision in untreatable eye conditions, such as retinitis pigmentosa and age-related macular degeneration. Our previous study suggested that ES promoted retinal function and the proliferation of progenitor-like glial cells in mice with inherited photoreceptor degeneration; however, the underlying mechanism remains obscure. Müller cells (MCs) are thought to be dormant residential progenitor cells that possess a high potential for retinal neuron repair and functional plasticity. Here, we showed that ES with a ramp waveform of 20 Hz and 300 µA of current was effective at inducing mouse MC proliferation and enhancing their expression of progenitor cell markers, such as *Crx* (cone–rod homeobox) and *Wnt7*, as well as their production of trophic factors, including ciliary neurotrophic factor. RNA sequencing revealed that calcium signaling pathway activation was a key event, with a false discovery rate of 5.33 × 10^−8^ (*p* = 1.78 × 10^−10^) in ES-mediated gene profiling changes. Moreover, the calcium channel blocker, nifedipine, abolished the observed effects of ES on MC proliferation and progenitor cell gene induction, supporting a central role of ES-induced Ca^2+^ signaling in the MC changes. Our results suggest that low-current ES may present a convenient tool for manipulating MC behavior toward neuroregeneration and repair.

## 1. Introduction

Müller cells (MCs) are the major type of glial cells in the retina [1]. They extend from the inner limiting membrane (ILM) to the outer limiting membrane (OLM) and play a crucial role in maintaining retinal homeostasis. MCs in lower vertebrates, such as zebrafish, can repair and regenerate damaged retinal cells by differentiating into progenitor cells [2]. Mammalian MCs, unfortunately, do not exhibit this regenerative behavior upon cellular injury.

Electrical stimulation (ES) has been suggested to affect neural stem cell fate and function in vivo [3]. Our recent study reported potential reprogramming responses of MCs following ES in a mouse model of inherited photoreceptor degeneration [4]. In line with this observation, many years ago Chow and team accidently discovered that low electric current could reverse the visual field and function loss in a safety assessment of an artificial silicon retina microchip in retinitis pigmentosa (RP) patients, in which the effect was observed far from the initial stimulation point [5]. That study provided evidence that low electrical current could have positive therapeutic effects on degenerative retinal diseases, and since then, animal and human studies have reported promising results. The use of electricity in medicine is not unperceivable [6], as it is widely used in cardiology, neurology, and psychiatry, gradually emerging in orthopedics and regenerative medicine as better and faster healing of wounds and bones [7,8,9,10,11,12]. Bioelectrical signaling has been studied closely and influences the epigenetics and transcriptional cascades that control cellular migration, proliferation, differentiation, and cell death [13]. Injury to rat corneas have also revealed different electrical current densities in wounded and healthy tissue that attracts or repels the cellular healing response [14]. Electricity is known to have a great effect on cellular healing power, but the use of this technology in ophthalmological diseases is still absent.

Clinical studies have shown that low current electrical stimulation improves vision and retinal function in several eye diseases [6], such as traumatic optic neuropathy (TON) [15], central or branch retinal arterial occlusion (CRAO or BRAO) [16], dry age-related macular degeneration (AMD) [12], and retinitis pigmentosa [17,18,19]. Emerging evidence suggests that low-current electricity increases the MC’s release of neurotropic factors, e.g., brain-derived neurotropic factor (BDNF) [20], ciliary neurotropic factor (Cntf) [21], fibroblast growth factor 2 (FGF-2) [22], and insulin-like growth factor-1 (IGF-1) [23]. These neurotrophic factors may have neuroprotective effects and promote the regeneration of damaged retinal tissue [24]. Other studies suggests that CNTF prevents photoreceptor apoptosis by stimulating MC production of photoreceptor survival factors [25]. Additionally, we know that electro stimulation (ES) enhances the ganglion cell survival rate in rats after axotomy and can increase the photoreceptor survival rate indirectly by improving retinal function, as seen in rabbits and mouse RP models [26]. Electricity also shows the ability to reduce inflammatory factors such as interleukin-1b (IL-1b) and tumor necrosis factor alpha (TNF-α) [21], which makes the clinical implementations almost impossible to ignore.

Unfortunately, there is insufficient or insufficiently strong evidence that can explain the vision improvements seen post-ES. It remains unclear how and why ES works on the retina as it does. It is still unknown if electricity actually alters MCs at the gene level, enhancing their regenerative potential, or if the different electrical parameters are influencing changes. The aim of the present study is to examine whether electricity can improve retinal glial cell regenerative abilities and functions. We believe that electricity might drive their genetic profile towards a more effective, protective, and reparative entity by encouraging MC transformation towards a progenitor-like phenotype. We also want to understand if different electrical conditions can be optimized to maximize the glial cells’ regenerative yield, which might provide some insight into why electricity actually improves vision.

## 2. Methods

### 2.1. Animals

Adult C57BL/6J mice were purchased from the Jacksons Laboratory, Bar Harbor, ME, USA. Postnatal day 6–7 (P6–7) mouse pups of mixed genders were used in all experiments for MC culture preparations. All animal experiments were performed in accordance with the protocols approved by the Institutional Animal Care and Use Committee of Schepens Eye Research Institute and followed the standards of the Association for Research in Vision and Ophthalmology. The mice were kept on a 12-h light/dark cycle with free access to food and water.

### 2.2. Isolation and Culture of Müller Cells (MCs)

We extracted and isolated retinas after euthanizing the postnatal day 6–7 (P6–7) as previously described [27]. Primary MCs obtained by this method were considered 95% pure, excepting the remaining cells to be microglia and possible endothelial cells [28]. The retinas were dissected in Dulbecco’s modified Eagle’s medium/F12 medium (DMEM/F12) solution and dissociated with 500 µL papain (Worthington Biochemical Corporation) for 10 min at 37 °C, with vigorous tapping after 5 min, according to the manufacturer’s protocol. Papain inhibitor was added to terminate the papain digestion; thereafter, the cells were centrifuged for 5 min with 340 relative centrifugal force (RCF). The cells were resuspended in defined medium, seeded on culture plates, and incubated at 37 °C with 5% CO_2_ and 95% humidity atmosphere. The defined medium consisted of DMEM/F12 with 10% fetal bovine serum (FBS) and 1.0% penicillin/streptomycin (P/S). The medium was changed three times a week. After 2 weeks of incubation (which provided the MCs sufficient time to mature in vitro), the cells were dissociated with trypsin (passaged) and counted with a Cellometer Auto T4 automated cell counter (Nexcelom, Lawrence, MA, USA), and seeded at 75,000 cells/well on poly-D-lysine (PDL, Millipore, Darmstadt, Germany; 10 µg/mL)-coated cover glasses in a 24-cell well plate. The cells were incubated for 1 day before ES treatment. After the treatment, the cells were transferred to wells containing 2.5 µM 5-ethynyl-2′-deoxyuridine (EdU)/medium solution for 48 h.

### 2.3. Labeling, Immunohistochemistry and Staining

To label the proliferating cells, we followed our published procedure [29]. Briefly, MC cultures were fixed in 4% paraformaldehyde (PFA) for 10 min and rinsed three times with 1% bovine serum albumin (BSA) in phosphate-buffered saline (PBS), followed by 20-min treatment with 0.3% Triton. A Click-iT cell reaction buffer kit (Thermo Fisher Scientific) was used to distinguish and label Edu+ proliferating cells according to the manufacturer’s protocol (Invitrogen). The primary antibody was rabbit glutamine synthetase (GS); the secondary antibody was anti-rabbit Cy3 (both, Jackson ImmunoResearch Laboratories). The cover glasses were stained with 4′,6-diamidino-2-phenylindole (DAPI) and mounted. Nine random areas were imaged from each cover glass, and the percentage of EdU+ among DAPI+ nuclei was calculated.

### 2.4. Electrical Stimulation (ES) of MCs In Vitro

An STG4000 stimulus generator (Multi Channel Systems, Reutlingen, Germany) was used to apply ES to the cells in DMEM. Eight wells of MCs were stimulated simultaneously by immersing platinum/iridium microelectrodes connected to the STG4000 unit; four wells received ES, and four wells received sham stimulation in a cell culture incubator with 5% CO_2_. Four parameters were used to establish the best proliferation condition, namely frequency, pulls per second (PPS) current (µA), pulse duration, and waveform. We began the optimization by establishing the best PPS, namely 10–100 PPS at a fixed current, rectangular waveform, with phase duration. Then, different currents (10–500 µA) with a fixed rectangular waveform, phase duration, and optimal PPS were applied. Next, different pulse durations were applied, namely 0.5–5 ms/phase, with optimal PPS, current, and a fixed rectangular waveform. Lastly, we investigated the waveform with optimal PPS, current, and pulse duration; we tested rectangular, ramp, and sinusoidal waveforms. All parameters were repeated with a fixed ramp waveform.

### 2.5. RNA Sequencing and Data Analysis

MCs were isolated from P6–7 mouse pups as previously described and harvested after 2 weeks’ incubation [27]. Each culture consisted of MCs derived from 3 to 4 pups, which was then divided into sham and ES treated groups. Cells of two independent cultures were combined to provide sufficient mRNAs for one pair (one ES-sample and one sham-treated sample) of RNA sequencing data. Thus, the results of RNA sequencing analysis represented data collected from MC cultures derived from a total 6 independent litters or >20 mouse pups that gave rise to 3 sham-control (Ctr) and 3 ES-treated groups. mRNA was extracted 48 h after ES- or sham-stimulation using an RNeasy Plus Mini kit (Qiagen, Venlo, the Netherlands) [30,31]. The samples were sequenced by Girihlet (Oakland, CA, USA), and RNA with an RNA integrity number (RIN) of 9 was reverse-transcribed to complementary DNA (cDNA). cDNA libraries were prepared using 500 ng total RNA using a TruSeq RNA sample preparation kit v2 (Illumina). mRNA sequencing and data analysis of cDNA libraries were sequenced on the Illumina NextSeq platform to obtain 80-bp single-end. The reads were trimmed, 2 nt (nucleotides) on each end, to remove low-quality parts and to improve mapping to the genome. The 78-nt reads that resulted were compressed by removing duplicates but keeping track of how many times each sequence occurred in each sample in a database. The unique reads were aligned to the mouse genome mm10. The raw data was uploaded to the NCBI site with the SRA accession number PRJNA608181. Misreads that crossed exon–exon boundaries, as well as reads with errors and single-nucleotide polymorphisms (SNPs)/mutations did not have substantial impacts on estimating the expression levels of each gene. Each mapped read was then assigned annotations from the underlying genome. In multiple annotations (e.g., a microRNA (miRNA) occurring in the intron of a gene), a heuristics-based hierarchy was used to provide each read with a unique identity. This was then used to identify the reads belonging to each transcript and establish coverage over each position on the transcript. This coverage was non-uniform and spiky; thus, we used the median of the coverage as an estimate of the gene’s expression value. The expression in different samples was compared using quantile normalization. The expression level ratios were then calculated to estimate the log (to base 2) of the fold change (FC). To prevent low-expression genes from dominating the list of genes with a large FC, we added a regularizer (5 or 10) to each value, ensuring that genes with expression around 5 or <5 would appear to have low FC. Enrichment analysis was performed by submitting genes with >1.5 FC between the ES and control groups to WebGestalt (http://www.webgestalt.org) [32] for Gene Ontology (GO) and Kyoto Encyclopedia of Genes and Genomes (KEGG) pathway enrichment analysis, in which mouse genome protein-coding genes were used as the reference gene list. There was a minimum of five genes per category, and over-representation enrichment analysis methods were applied. Categories with false discovery rates (FDR) < 0.05 were considered significant. Volcano plots were plotted with R software. For the volcano plots, the paired *t*-test *p*-value and FC were −log10- and log2-transformed, respectively. Genes with FC > 1.5 were marked with a red dot. For the heatmap, the heatmap.2 function was used, in which the gene expression matrix was log2(value+1)-transformed. A.I.R. (Artificial Intelligence RNA Seq (RNA sequencing), transcriptomics.sequentiabiotech.com) was used as a supplement.

### 2.6. Calcium Channel Blocker and MC Proliferation

MCs were isolated as described above and seeded on a PDL-treated cover glass at 75,000 cells/cover glass for 24 h. The MCs were then treated with optimal ES or sham conditions for 1 h. Subsequently, the MCs were incubated for 48 h in DMEM/F12 containing 2.5 µM EdU and 1 µM nifedipine (an L-type calcium channel blocker) [33] in defined medium for MCs. By immersing the MCs in calcium channel blocker post ES for 48 h, we aimed to parallel the timing of RNA-sequencing analysis and evaluate the long-term impact of ES on MC proliferation and gene transcription. The MCs were processed for EdU labeling, as described above, and for RNA extraction.

### 2.7. Real-Time PCR

We used a Qiagen RNeasy Plus Mini kit to extract mRNA. The mRNA was converted to cDNA using a SuperScript IV First-Strand Synthesis System (Invitrogen) according to the manufacturer’s instructions. Appendix A lists the primers. We then performed real time-PCR (RT-PCR) using a KAPA SYBR FAST kit (Kapa Biosystems), and the experiments were performed in triplicate. The expression level of GAPDH (glyceraldehyde-3-phosphate dehydrogenase) was used as the internal control.

### 2.8. Statistical Analysis

The data were analyzed with GraphPad Prism 5.0 by different statistical methods, including paired and unpaired t-tests and one-way-ANOVA, and the method of false discovery rate (FDR) was used for RNA sequencing analysis. Data were presented as mean ± S.E.M, and *p* < 0.05 was designated as statistically significant.

## 3. Results

### 3.1. ES Promotes MC Proliferation

Zebrafish MCs can spontaneously repair and regenerate their retinas, but those of mammals cannot. For this to happen, mammalian MCs must first be able to proliferate and increase their repair and regenerative abilities exponentially. Therefore, we investigated the optimal ES conditions for improving glial cell proliferation in vitro. MCs were isolated as previously described [27,34]. Proliferation was assessed using the EdU incorporation assay [29]; the MC purity was calculated to be ~95% as reported, and as confirmed by counting the cells immunolabeled positive for the MC marker, glutamine synthetase (GS) (Figure 1A).

MCs were treated with low-current electricity with a rectangular waveform at different frequencies, phase durations, and current amplitudes. The rectangular waveform was initially selected because it is most commonly used in all reported animal and clinical studies and is most consistent with the natural electrical signal of neurons. MCs were initially stimulated for 1 h with standard 100 µA, a rectangular waveform, at 1 ms/phase duration and 10–100 PPS. We found that 10 and 20 PPS marginally upregulated MC proliferation by ~1.2-fold (Appendix A). Second, we explored the optimal phase duration by varying between 0.5 and 5.0 ms/phase under 20 PPS, 100 µA, and a rectangular waveform. While ES of 1 ms/phase duration tended to lead to the highest number of proliferating MCs, we detected no significant differences among groups with various phase durations (Appendix A). Third, we investigated the optimal current amplitudes by varying between 10 and 500 µA under 20 PPS, 1 ms/phase duration, and a rectangular waveform. We found that 300 µA yielded optimal MC proliferation (Appendix A), but no statistical significance was detected at this timepoint. The data suggest that a rectangular waveform may not be as optimal for stimulating MC as has been shown to benefit retinal ganglion cells [34].

Next, we explored the different ES waveforms: rectangular, sinusoidal, and ramp (saw-tooth) (Figure 2A). At a fixed frequency of 20 PPS and 100 µA current amplitude, we noted that ramp waveforms significantly improved MC proliferation by nearly 2-fold compared to the non-stimulated control group and the sinusoidal or rectangular waveform (*p* < 0.005, paired *t*-test) (Figure 2B). To optimize the ES conditions, we performed a re-run of the current and frequency with a fixed increasing-ramp waveform. Under a fixed 20 PPS, currents of 100 µA and 300 µA with a ramp waveform stimulated significantly increased MC proliferation by 1.6-fold over the non-stimulated control group (paired *t*-test, *p* < 0.05) (Figure 2C). To further verify these conditions, we next used a 300 µA current and a ramp waveform and confirmed that the frequency of 20 PPS was optimal for improving MC proliferation compared to the non-stimulated controls (Figure 2D). Moreover, there was no significant differences among the peak values of MC proliferation under the optimal ES condition in all experimental groups (Figure 2B–D). These results indicate that ES at 20 PPS, 100–300 µA, and a ramp waveform has the potential to significantly enhance MC proliferation in vitro.

### 3.2. RNA Sequencing Gene Profiling and Analysis of MCs

To exploit ES-induced gene expression changes in MCs, purified MCs were stimulated with optimal ES condition as indicated above (20 PPS, 300 µA, ramp waveform) for 1 h in vitro and incubated for 48 h before being processed for RNA sequencing analysis. Thus, cells were provided with sufficient time for induction of mRNA expression. The differences in gene expression between the ES and control groups were evaluated with quantile normalization and a paired *t*-test. The fold change cut-off was set to >1.5 with FDR < 0.05 and *p* < 0.05 for each gene. This dataset yielded 479 differentially expressed genes (DEG) (Figure 3A). The samples were analyzed with ingenuity pathway analysis (IPA, Qiagen), GO, KEGG, distant regulatory elements (DiRE), and supplemented with A.I.R. The volcano plots illustrate the relationship between the FC and significant gene expression (Figure 3A). 

We first performed DiRE analysis to determine the overall changes of ES-induced transcription factors (TFs), which are key regulatory elements governing gene expression, such as enhancers, repressors, and silencers [35]. Based on the enhancer identification (EI) method, DiRE mapped out 188 potential regulatory elements (REs) (Figure 3B). The top three ES-induced TF binding sites (TFBS) included TATA, NKX6.2, and CEBP (Figure 3C), all of which have been implicated in mediating trans-factors for driving neurogenesis and regeneration [36,37,38], such as *Notch/Hes* signaling [39,40]. GO analysis further pointed out that ES indeed downregulated *Notch/Hes1* (hes family bHLH TF 1), crucial TFs that regulate progenitor cell fate and neurogenic potential [41]. We verified this finding by quantitative real-time PCR (qPCR) and confirmed the significant downregulation of *Notch3* and *Hes1* in ES-treated MC cultures as compared to the controls (Figure 3D). Pathway analysis of *Notch/Hes1* downregulation predicted that ES drives MCs toward a progenitor and photoreceptor cell fate (Figure 3E) [41]. 

To study the biological pathways and gene networks activated by the ES, we employed KEGG analysis. This linked ES treatment to the calcium signaling and synaptic activities. KEGG analysis revealed prominent upregulation of the calcium signaling pathway (FC > 1.5, FDR = 5.33 × 10^−8^ (*p* = 1.78 × 10^−10^) (Figure 4A)) and the voltage-gated calcium channels, the pathway of which acts directly or indirectly to mediate cell growth and development. Among the 62 of 183 genes involved in the calcium signaling cascades, a total of 28 genes, including *Camk2b* (calcium/calmodulin-dependent protein kinase II beta) and *Ptk2b* (protein tyrosine kinase 2 beta), which are important for synaptic formation and neuroplasticity, were upregulated. The MAPK–ERK (mitogen-activated kinase–extracellular signal-regulated kinase) pathway that is crucial for cellular proliferation, development, and differentiation, was upregulated (Figure 4B), consistent with the observed MC proliferation post-ES. As the calcium signaling pathway was clearly over-represented, several synaptic signaling pathways, such as the glutamatergic and GABAergic signaling pathways, were also over-represented in the ES-treated samples (Figure 4A), with FC > 1.5, in comparison to the reference *Mus musculus* genome glutamatergic synapses (FDR = 9.84 × 10^−5^; *p* = 9.87 × 10^−7^) and GABAergic synapses (FDR = 0.00533, *p* = 0.000179). These changes support that ES induces membrane depolarization, likely involving voltage-gated channel activation and leading to Ca^2+^ influx and sensitization for certain neurotransmitters and synaptic signaling. An overarching look at ES-activated signaling events suggests that ES confers MC with predictively increased cell growth capacity (e.g., MAPK/ERK and CREB), while it weakly predicted inhibition of NFATc (nuclear factor of activated T cells 1) or the immune/inflammatory pathways (Figure 4B). This altered MC genetic landscape likely provides them with higher plasticity and progenitor cell/neurogenic potential.

### 3.3. The L-Type Calcium Channel Mediates ES-Induced MC Proliferation and Differentiation

To corroborate the finding from the RNA-sequencing analysis, we tested the effects of calcium signaling on ES-mediated MC proliferation in cultures by administration of L-type calcium blocker nifedipine. Several isoforms of the L-type calcium channels exist in mice, and previous studies showed that dihydropyridines (nifedipine), a standard pharmacological compound (L-type calcium channels blocker), blocks IGF-1 production in MCs post-ES. It is believed to work by obstructing the high voltage-activated L-type calcium channel in rodent MCs [28,42]. As expected, ES significantly increased MC proliferation. In contrast, nifedipine significantly attenuated the ES-induced proliferation similarly to that in the sham control and nifedipine-only groups (Figure 5A), supporting that the L-type calcium channel is critically involved, and its blockade is sufficient to abolish ES-induced effects on MC proliferation. 

To further determine if the calcium-mediated pathways are keys to the ES-induced neurogenic potential of MC, we investigated the effect of nifedipine on the expression of ES-induced retinal and photoreceptor progenitor markers previously reported [29], which include Crx, Wnt7, Chx10, Sox2, and Pax6, as well as trophic factor-related genes. Following an optimal ES treatment (20 PPS and 300 µA ramp) in vitro, as expected we observed significant increases of the progenitor cell markers *Crx* (cone–rod homeobox) and *Wnt7* following 1-h stimulation (Figure 5B). This data was consistent with the RNA sequencing results, which revealed a 35-fold increase of Crx mRNA levels following ES. Upregulation of the photoreceptor progenitor marker *Crx* reached 8-fold. Significant upregulation of these genes was not observed at 15 min post-ES (Appendix A). Moreover, CNTF, a key trophic factor that supports photoreceptor differentiation and survival, was also found to be upregulated nearly 3-fold after ES. These data are consistent with the notion that ES suppresses *Notch*/*Hes1* signaling and promotes MC trans-differentiation toward a photoreceptor progenitor cell fate.

## 4. Discussion

We have shown that low electrical currents with 20 Hz, 100–300 μA current, and a ramp waveform represent the optimal ES conditions for promoting MC proliferation in vitro. Accompanied with it, MCs alter their ability to express progenitor cell markers. Activation of the calcium signaling is shown to be a key event underlying these changes, and by blocking the voltage-gated calcium channels, the ES-induced effects on MC were attenuated. Optimal ES parameters improved the regenerative potential and function of MC by driving the gene profile towards an effective, protective, and reparative nature. These effects of ES were attenuated by nifedipine, L-type calcium channel, which reduced the expression of the progenitor-related genes *Wnt3*, *Wnt7*, and the photoreceptor progenitor-related genes *Crx* and *Chx10* (visual system homeobox 2) in the ES-treated MCs. Moreover, nifedipine abolished the effects of ES-induced MC proliferation.

A strength of the current studies regards the employment of isolated MCs, which enables evaluation of the direct effect of ES on MC gene profiling and cellular behavior, such as survival and proliferation, without being interfered by other cell types, whereas, at the same time, this also presents a critical limitation because the observed results may not correlate precisely to those in vivo. While this limitation is inevitable for all culture studies, the results provide important clues for the underlying complex ES mechanism. Future studies are needed to further verify these findings, especially the gene expression changes in MCs after ES, in in vivo models. 

Several factors and pathways are critically involved in the ES-induced MC responses, as our sequencing data suggest that electricity increases the cellular glial expression of synaptic-related genes, which are crucial for neuron–cell communications. These changes are likely to increase MC sensitivity to neurotransmitters, which increase the neuronal activities between the MC and the bipolar, amacrine, and retinal ganglion cells. It is likely that the visual improvements observed post-ES will take place as long as the compensatory mechanism triggered by ES is at play, as seen in several animal models and clinical trials supporting this notion [12,19,43]. The RNA sequencing data further predicted that the calcium signaling pathway might play a significant role in ES-induced actions, and the literature states that rodent retina expresses the following three subunits of the voltage-gated L-type calcium channel [44]: Ca_v_1.2, which is seen in all retinal cells, including MCs, while Ca_v_1.3 and Ca_v_1.4 are primarily expressed in retinal neurons, with Ca_v_1.4 expressed exclusively in photoreceptor synaptic terminals [45]. Other groups have shown that ES-induced growth factor secretion (IGF-1) from MC can be blocked by nifedipine, a voltage-dependent calcium channel blocker [28]. The present study is in agreement with this finding, showing the loss of MC proliferation and progenitor cell gene expression by blocking the calcium channel activities. Our findings suggest that Ca^2+^ signaling plays a central role in the underlying mechanism of ES.

We found significant upregulation of the progenitor cell markers, such as *Wnt7*, *Crx*, and CNTF in MC after 1-h ES treatment. It was reported that *Wnt* drives the proliferation of a subpopulation of MCs [46] and increases the number of progenitor-like MCs [47]. *Wnt* may stimulate MC proliferation through activation of the *Wnt*–*Lin28*–*let7* pathway, as it was seen in the injury models [48,49]. It is likely that when MCs exhibit progenitor-like cell properties, they function as a transient intermediary entity that supports the survival and function of retinal cells to improve vision [50]. The increased progenitor cell markers can also be explained in part by the ES-induced downregulation of *Notch1* and *Hes1* in MCs. Continuous expression of HES1 initiates MC proliferation [41], while downregulation of the notch–HES pathway alters the MC fate by promoting cell transdifferentiation and neuroregeneration [41]. Upregulation of CNTF itself may improve vision, as has been shown in patients with RP, possibly by stimulating MC to produce photoreceptor survival factors [25], leading to reduced apoptosis and improved photoreceptor morphology [51]. Moreover, increased calcium signaling may also link ES-induced MC proliferation through different pathways such as ERK–MAP signaling. Much has been discovered over the years in the field of bioelectrical signaling, as even the smallest disruption of electrical current in the cornea can attract or repel cells, but MCs remain inadequately understood [13,14].

The fact that ES alters the genetic landscape in MCs to enhance their progenitor cell potential is remarkable. Our RNA sequencing analysis represented data collected from three independent MC cultures taken from a total of six independent litters or >20 mouse pups. The results support the improved potential of ES-treated MCs to regenerate neurons under a cell injury response. As with all therapies, one of the major challenges regarding the use of ES is to determine the appropriate stimulus strength or dosing. While most studies focus on optimizing the electroceutical dosing with regard to current, duration, and frequency, little has been done to discover optimized waveform shapes. Evidence has emerged where, under many circumstances, non-rectangular waveforms, e.g., ramp waveforms, can evoke stronger responses at a given charge, indicating higher charge efficiency, thus leading to enhanced clinical efficacy [52,53]. Our observation in MC cultures also supports the premise that continuous ramp waveforms may be more effective for activating cell signaling in MCs. The fact that ES conditions at 20 Hz and 100–300 µA yielded optimal MC proliferation was unsurprising, as they are in agreement with other reports that the similar parameters had positive effects on different retinal cell types [54,55,56]. 

Other factors that also possibly influence the MC after ES treatment have not been addressed, such as how the medium responds to electricity and if hydrolysis or oxidized components can drive the changes observed post-ES. The ES effect on MC membrane potential is also unknown, and more electrophysiological experiments are required. It is important to acknowledge that the glial cell population of 95% MCs may contain small remnants of other cell types such as microglia and endothelial cells, which can contribute to the cellular effect observed post-ES. The reduction of microglia activity alone can provide a positive ES effect, as reported previously [57]. As the field of electroceuticals continues to advance, we anticipate that it will open new doors and treatments with other effective waveforms and ES conditions. The observed changes are subtle, but the present study and future studies are needed to fully characterize and understand the genes and mechanisms at play.

## 5. Conclusions

Low electric current, especially with a ramp waveform, improves MC proliferation and promotes their progenitor cell potentials. Based on the genetic and cellular responses observed, our results suggest that ES can have a direct impact on glial cells (MCs), driving toward neuroregeneration and repair. Low current ES can be a convenient, noninvasive tool in regenerative medicine when applied correctly. Exactly how electricity affects the whole retina remains to be elucidated, but it is most promising to realize that ES not only modulates neuronal activity, but can also be tuned to act directly on MCs to modulate their responses under disease or injury conditions.

## Figures and Tables

**Figure 1 cells-09-00781-f001:**
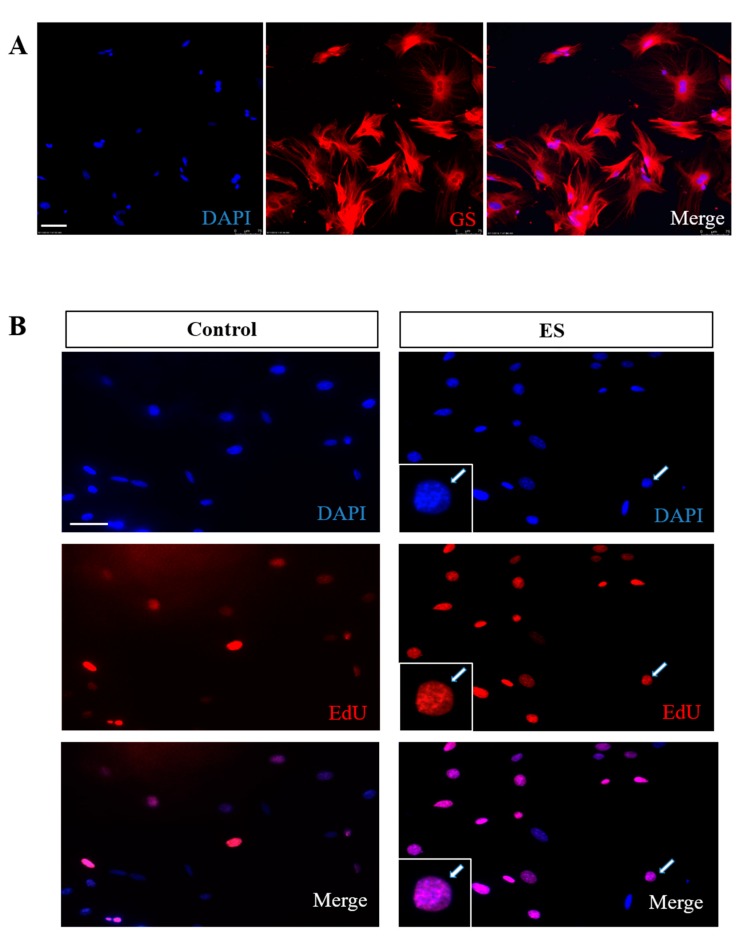
The effect of electrical stimulation (ES) on cultured Müller cells (MCs). (**A**) Photomicrographs showing isolated MC cultures immunolabeled for MC specific marker glutamine synthetase (GS; red) and counter-stained with a nuclear marker 4′,6-diamidino-2-phenylindole (DAPI) (blue). Note that most cells in isolated MC cultures expressed typical MC marker GS. (**B**) MC proliferation assessed by EdU incorporation. MCs were stimulated with ES or sham-treatment (control) for 1 h. Note the increased number of EdU+ (red) cells 2 days after ES treatment as compared to the control group. Scale bar = 20 µm.

**Figure 2 cells-09-00781-f002:**
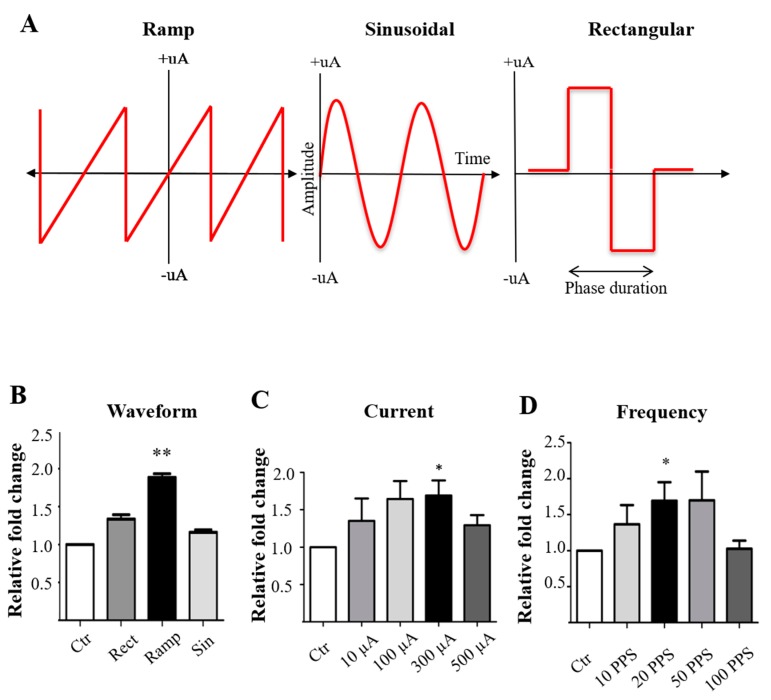
Optimal conditions of ES with a ramp waveform. (**A**) The three waveforms used in the study: ramp waveform with negative and positive ramp, the typical sinusoidal waveform with current fluctuations in a typical sinusoidal pattern, and the biphasic rectangular waveform. (**B**) Changes in MC proliferation following ES of different waveforms at 20 PPS and 100 μA. Note that ramp waveform significantly promoted MC proliferation by ~2-fold as compared to the control group. (**C**) Changes in MC proliferation when stimulated with ramp waveforms of ES at 20 PPS and different current levels. Paired *t*-test, *p* < 0.05 compared to controls. (**D**) Changes in MC proliferation with ES of fixed 300 µA current and ramp waveform at various frequencies. Note that 20 PPS yielded significantly increased MC proliferation. Value = means ± S.E.M. * *p* < 0.05, ** *p* < 0.01 compared to controls by paired *t*-test (*n* = 5/group).

**Figure 3 cells-09-00781-f003:**
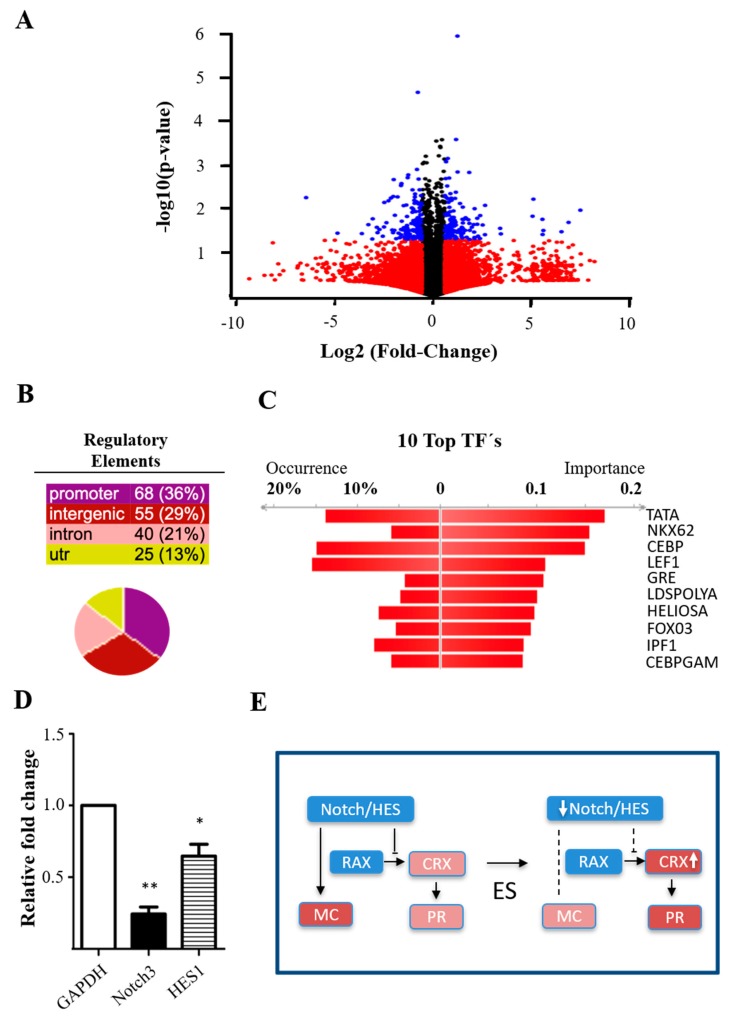
ES-induced transcription factor (TF) changes in cultured MC. (**A**) Volcano plot of differentially expressed genes (DEGs) results between ES- and control-treated MC groups. The y-axis corresponds to the *p*-value showing at −log10 of the paired *t*-test; the x-axis shows the fold change calculated at log2 value. The red dots represent the genes with expression changes <1.5-fold; the blue dots represent genes with expression change >1.5-fold and FDR and *p* < 0.05. (**B**) Summary of regulatory elements (REs) detected by distant regulatory elements (DiRE) analysis, which is represented relative to the input genes and categorized as promoter, intronic, intergenic, or UTR elements. (**C**) Graphical representation of candidate RE scores to the list of most important TFs. The “occurrence” represents the fraction of putative REs that contain a particular TF binding site, and the “importance” is defined as the product of the TF occurrence and its weight. (**D**) Results of qPCR confirming the significant downregulation of *Notch3* and *Hes1,* 48 h after ES in purified MC cultures (* *p* < 0.05 and ** *p* < 0.01, compared to the controls by paired *t*-test; *n* = 4/group). (**E**) Schematic illustration of the notch/HES signaling on MC fate determination.

**Figure 4 cells-09-00781-f004:**
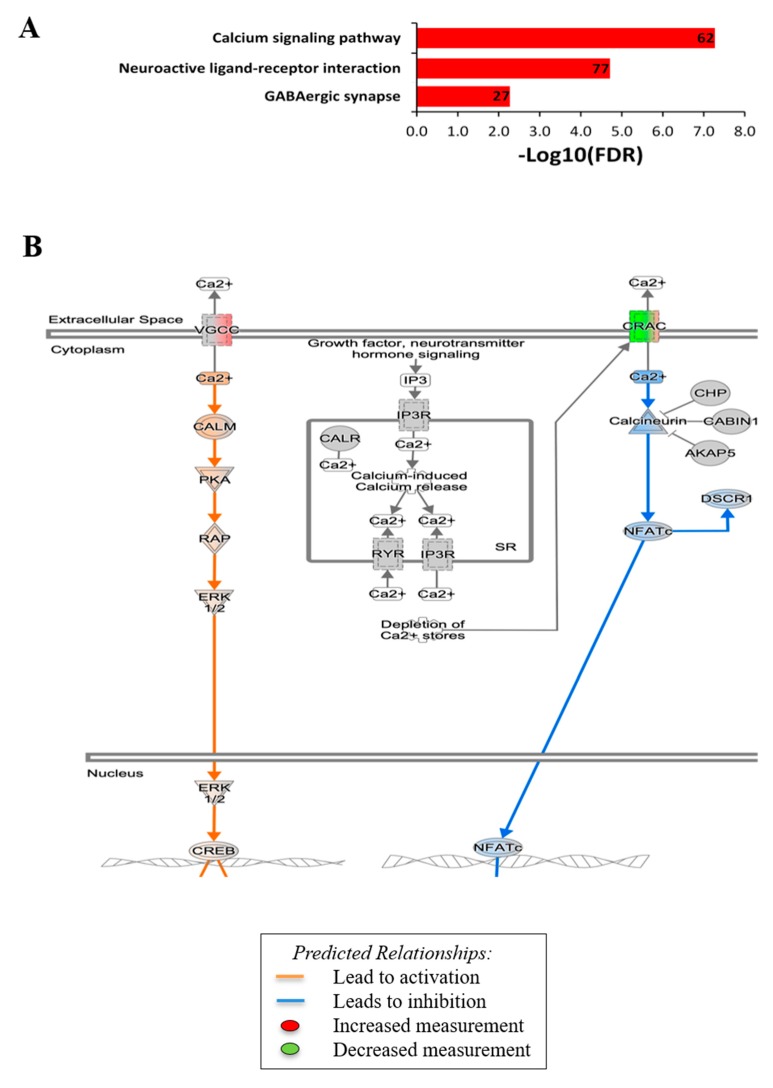
Upregulation of calcium signaling and cell growth events in MCs by ES. (**A**) Kyoto Encyclopedia of Genes and Genomes (KEGG) analysis showing the top three, particularly Ca^2+^-mediated signaling pathway, identified with a high number of genes changed upon ES-treatment (showed as numbers on the bars). (**B**) Schematic summary of ES-induced signaling pathway changes as shown by KEGG pathway analysis of all DEGs showing significant upregulation of the calcium signaling pathway and cell growth while inhibition of NFATc pathway with overlaid predictions. Orange lines predict activation and blue lines indicate inhibition of the downstream signal.

**Figure 5 cells-09-00781-f005:**
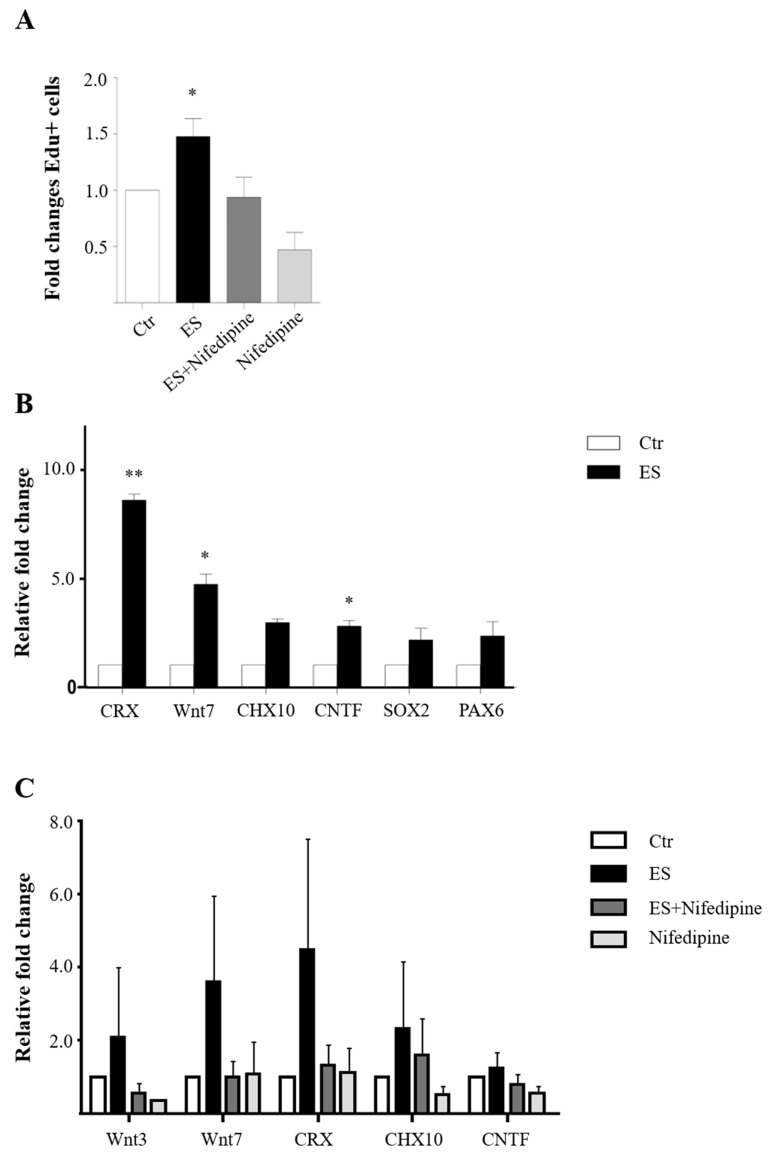
Essential involvement of calcium for ES-induced MC proliferation. (**A**) Improved MC proliferation after ES and the loss of this effect in the presence of 1 µM nifedipine (* *p* < 0.05, paired *t*-test). (**B**) Results of qPCR showing upregulation of progenitor cell markers and the neurotrophic factor *Cntf* in cultured MC at 48 h after ES with ** *p* < 0.005 and * *p* < 0.05 compared to sham treated control with paired *t*-test. (**C**) Nifedipine blocked ES-induced upregulation of progenitor markers such as *Wnt3*, *Wnt7*, *Crx*, and *Chx10* in cultured MC.

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
