# Peer review of "Electrical Stimulation Induces Retinal Müller Cell Proliferation and Their Progenitor Cell Potential"

_cells, 2020, doi:10.3390/cells9030781_

Round 1

Reviewer 1 Report

The manuscript by Enayati, et al., describes the use of non-invasive electrical stimulation promotes Müller glia cell proliferation and expression of progenitor cell markers. Further, calcium signaling pathways were implicated in this process. Overall, the concept is novel and of interest to both the vision field, in general, and the regenerative medicine field more broadly. However, a few concerns need to be addressed:

  1. For the calcium channel studies, why was the blocker added after electrical stimulation? It seems more appropriate to add prior to ES.
  2. Three replicates for RNA-Seq studies is a major limitation of this study. It is likely that more differentially expressed genes would have been identified with more replicates, which could possibly change the results. The reason being that small variation has a larger effect on fewer replicates. This should be addressed by adding more replicates, or stating in the manuscript this limitation.
  3. How was RNA-Seq raw data aligned? Which mouse genome and transcriptome build were used?
  4. The raw RNA-Seq data needs to be submitted to a repository. Given the NIH funding stated in the acknowledgments, this should be in SRA or GEO. An accession number should be made available to the reviewers prior to accepting the manuscript for publication.
  5. Was Crx and Wnt7 observed to be differentially expressed in the RNA-Seq data? If so, this should be included in the manuscript, as it seems random as to why the authors chose to look at these genes and not other progenitor markers.

Author Response

Comment: 1. For the calcium channel studies, why was the blocker added after electrical stimulation? It seems more appropriate to add prior to ES.

Response:  We thank for the reviewer’s comment and have now clarified in the method section: By immersing the MC´s in calcium channel blocker post ES for 48 hours, we aimed to parallel the timing of RNA-seq analysis and evaluate the long-term impact of ES on MC proliferation and gene transcription.  (Page 4, line 172-174)

Comment: 2. Three replicates for RNA-Seq studies is a major limitation of this study. It is likely that more differentially expressed genes would have been identified with more replicates, which could possibly change the results. The reason being that small variation has a larger effect on fewer replicates. This should be addressed by adding more replicates, or stating in the manuscript this limitation.

Respond: We appreciate the Reviewer’s comment. We have now clarified: “Each culture consisted of MC derived from 3-4 pups, which was then divided into sham- and ES-treated groups. Cells of two independent cultures were combined to provide sufficient mRNAs for one pair (one ES-sample and one sham-treated sample) of RNAseq data. Thus, the results of RNAseq analysis represented data collected from MC cultures derived from a total 6 independent litters or >20 mouse pup that gave rise to 3 sham-control (Ctr) and 3 ES-treated groups.”  (Page 3, lines 131 – 135 and Page 12, lines 388 - 389).

Comment: 3. How was RNA-Seq raw data aligned? Which mouse genome and transcriptome build were used?

Respond: We have now added in the manuscript the data provided to us by our company: The unique reads were aligned to the mouse genome mm10, (Page 4, Line 144) and (Page 3-4, line 141-157).

Comment: 4 The raw RNA-Seq data needs to be submitted to a repository. Given the NIH funding stated in the acknowledgments, this should be in SRA or GEO. An accession number should be made available to the reviewers prior to accepting the manuscript for publication.

Respond: The raw RNA-seq data have now been uploaded to the NCBI site with the SRA accession number of PRJNA608181 (Page 4, lines 144 – 145).  

Comment: 5. Was Crx and Wnt7 observed to be differentially expressed in the RNA-Seq data? If so, this should be included in the manuscript, as it seems random as to why the authors chose to look at these genes and not other progenitor markers.

Respond: We have now clarified in the Result section: “To further determine if the calcium-mediated pathways are keys to the ES-induced neurogenic potential of MC, we investigated the effect of nifedipine on the expression of ES-induced retinal and photoreceptor progenitor markers previously reported, that include Crx, Wnt7, Chx10, Sox2 and Pax6, … This was consistent with the RNA-seq results, which revealed 35-fold increase of Crx mRNA levels following ES.” (page 10, line 319-324).

Reviewer 2 Report

Please see my comments below:

1) Line 58: small clinical studies? it is okay to have small clinical studies if they were designed and had enough power to detect differences. My question is that were these small clinical studies high quality?

2) Line 83: please include location for the laboratory.

3) For the methods, please cite appropriate references so that the readers can refer to these methods in future. In addition, the methods were also unclear in some parts. For example, how mouse pups of mixed sex in each group were used. These were not explained in detail.

4) line 177: What were the other statistical methods used? Please list them.

5) Figure 2: how about between-group comparison? Were they any significant differences?

6) Similar questions were also for other figures: how about between-group comparison? Were they any significant differences?

7) Were the results adjusted for any covariates?

8) references were not formatted correctly to the journal citation style.

9) please include a paragraph in the discussion for strengths and limitations.

Author Response

Comment: 1. Line 58: small clinical studies? it is okay to have small clinical studies if they were designed and had enough power to detect differences. My question is that were these small clinical studies high quality?

Respond: We thank for the reviewer’s point and have now carefully checked the citations and removed the low quality or case studies that did not provide sufficient power. 

Comment: 2. Line 83: Please include location for the laboratory.

Respond: The location of the laboratory information has now been added into the method section: Bar Harbor, ME USA (Page 2, line 82)

Comment: 3. For the methods, please cite appropriate references so that the readers can refer to these methods in future. In addition, the methods were also unclear in some parts. For example, how mouse pups of mixed sex in each group were used. These were not explained in detail.

Respond: We have gone through the references cited in the Methods section and made corrections accordingly. In addition, we have now clarified: Postnatal day 6–7 (P6-7) mouse pups of mixed genders were used in all experiments of MC culture preparations (Page 2, lines 83 - 84).

Comment: 4. Line 177: What were the other statistical methods used? Please list them

Respond: We have now addressed this comment by including all statistical methods used, including Paired and unpaired t-tests and one-way-ANOVA, and the method of false discovery rate (FDR) was used for RNA-seq analysis. Data were presented as mean ± S.E.M, and P < 0.05 was designated as statistically significant (Page 4, lines 184-186).

Comment: 5. Figure 2: how about between-group comparison? Were they any significant differences?

Respond: We find significant differences between groups and have now clarified this in the text: At the fixed frequency of 20 PPS and 100 µA current amplitude, we noted that ramp waveforms significantly improved MC proliferation by nearly 2-fold compared to the non-stimulated control group and the sinusoidal or rectangular waveform (P < 0.005, paired t-test) (Fig. 2B) (Page 6, lines 219 – 221).

Comment: 6. Similar questions were also for other figures: how about between-group comparison? Were they any significant differences?

Respond: We performed one-way-ANOVA tests for between-group comparisons that included Fig. 2B-D and Fig. 5A and found no significant differences in the peak value of MC proliferation under the optimal ES conditions of the ramp waveform. This has now been clarified in the revised manuscript (Page 6, lines 227 – 229).

Comment: 7. Were the results adjusted for any covariates?

Respond: We thank the reviewer for the question. All of our results have been normalized to the control group and adjusted for covariates.  

Comment: 8. References were not formatted correctly to the journal citation style.

Respond: This has now been addressed.

Comment: 9. Please include a paragraph in the discussion for strengths and limitations.

Respond: We have now added a paragraph in the Discussion section to address the strengths and limitations: A strength of the current studies regards the employment of isolated MC, which enables evaluation of the direct effect of ES on MC gene profiling and cellular behavior, such as survival and proliferation, without being interfered by other cell types. Whereas, at the same time, this also presents a critical limitation because the observed results may not correlate precisely to that in vivo. While this limitation is inevitable for all culture studies, the results provide important clues for the underlying complex ES mechanism. Future studies are needed to further testify these findings, especially the gene expression changes in MC after ES, in in vivo models (Page 11, lines 347 – 353).

Round 2

Reviewer 1 Report

The authors have addressed my concerns. A very minor spell check is needed. For example, 'RNA seq' and 'RNA-seq' are used in the manuscript.

Reviewer 2 Report

Overall, the authors have improved the manuscript significantly. However, I can still spot some careless typo errors such as in line 184, "Pairs" should be "pairs". Please correct the grammatical mistakes throughout the manuscript.
